# Comparing two protocols of shock wave therapy for patients with plantar fasciitis: A pilot study

**Fatima A. L. Kalbani**[1], **Reime Shalash**[1,2], **Raneen Qadah**[1,2], **Tamer Shousha**[1,2,3,4,5]*

1 Department of Physiotherapy, College of Health Sciences, University of Sharjah, Sharjah, UAE,
2 Research Institute of Medical and Health Sciences, Neuromusculoskeletal Rehabilitation Research Group, University of Sharjah, Sharjah, UAE, 3 Faculty of Physical Therapy, Department of Physical Therapy for Musculoskeletal Disordered and its Surgery, Cairo University, Cairo, Egypt, 4 University of Sharjah Center of Excellence for Healthy Aging, Sharjah, United Arab Emirates, 5 Research Institute of Medical and Health Sciences, Healthy Aging, longevity, and Sustainability Research Group, University of Sharjah, Sharjah, UAE

☍ These authors contributed equally to this work.
* tshousha@sharjah.ac.ae

**Data Availability Statement:** All relevant data are within the paper and its Supporting Information files.

**Funding:** The author(s) received no specific funding for this work.

## Abstract

### Objectives

This pilot study primarily aimed to detect the adherence as well as the effect size required to estimate the actual sample size needed for a larger scale study to compare and evaluate the effectiveness of two extracorporeal shock wave therapy (ESWT) protocols along, with a physical therapy program in reducing pain and improving function among patients suffering from plantar fasciitis. The study also aimed to report the effects of the ESWT protocols used on pain and function.

### Methods

A total of 26 participants took part in the study, including 17 females and 9 males. The average age of the participants was 34 years with a body mass index (BMI) of 23 kg/m$^2$. Participants were divided into three equal groups; Group A received ESWT at a frequency of 15 Hz and intensity of 3, Group B received ESWT at a frequency of 10 Hz and intensity of 4, while Group C underwent the selected physical therapy program along with sham shock wave therapy as a control. Pain levels were assessed using the Visual Analog Scale (VAS) while functional improvements were evaluated using the Foot Function Index (FFI). Data was collected prior to treatment, after three sessions and at the end of six weeks (after six sessions).

### Results

The three groups were well matched, and the results revealed high adherence rates (90%, 90% and 80% respectively). Results also indicated reductions in pain levels and improvements in function for both intervention groups when compared to the control group. Group A demonstrated better outcomes compared to Group B while Group C showed relatively less improvement.

**Competing interests:** no competing interests exist.

## Conclusion

The study concluded a high adherence rate for the three groups as well as a small effect size detected of 0.282 that would suggest a total of 123 participants to be required to replicate the study on a larger scale. With regards to the findings of this pilot, the combination of ESWT and a targeted physical therapy program revealed a possible effective therapeutic approach for plantar fasciitis, with a higher frequency potentially yielding more favourable results.

## Introduction

Plantar fascia is a tissue band that runs along the bottom of the foot. It connects to each toe and is attached to the heel bone [1,2]. It provides the arch of the foot strength and support. When this tissue band gets strained or irritated, plantar fasciitis occurs [1]. Plantar fasciitis is a common musculoskeletal injury that cause heel pain, with symptoms of stabbing, non-radiating pain in the early morning of the proximal medio-plantar surface of the foot; the pain becomes worse at the end of the day [2].

[1,3]. It affects individuals across ages and activity levels [3]. It is considered a degenerative pathology rather than a primary inflammatory disorder [3]. The pain onset is sharp, and it may either decrease or become duller after light exercises/ activities [1].

The problem origin of the plantar fasciitis is usually due to anatomical anomalies of the foot, which cause biomechanical stress on the joints and supporting soft tissue structures, which fail to adjust due to either standing for a long time and repetitive nature of such demands or supraphysiological loads on them [4]. The human foot must perform two important functions: providing propulsive force in the latter part of the stance phase and absorbing the impact of the body weight in the early part [4]. This demands the foot to be flexible and soft when weight bearing, and rigid when pushing off [4].

Treatment options include minimally invasive treatments like platelet-rich plasma injections or steroids, non-invasive physical therapy modalities, exercise programs, and newer modalities such as shock-wave therapy [5,6]. If the patient didn't response to any of these treatments, then surgery might be necessary [3]. Treatment options for this condition may include combining several modalities, and gradually increasing the intensity of the treatment given [3]. A meta-analysis of randomized control trials has highlighted the effectiveness of shock-wave therapy and ultrasound in the treatment of planter fasciitis, where the visual analogue scale has improved more in the group that received shock-wave therapy rather than ultrasound, suggesting that it can be a superior alternative for plantar fasciitis treatment [6].

A meta-analysis evaluated the effectiveness of extracorporeal shock wave therapy in treating chronic plantar fasciitis and found that a moderate- and high- intensity of it, is effective in treating chronic plantar fasciitis [7]. Moreover, a systematic review and network meta-analysis study compared the effectiveness of focused and radial shock wave therapy in treating plantar fasciitis and concluded that when using focused shock wave therapy for plantar fasciitis, it is best to set the energy output at the highest and most tolerable within the medium intensity ranges and that radial shock wave therapy is considered a good alternative due to its lower price and probably better effectiveness [8].

As there were different sample and effect sizes reported in previous studies, the purpose of this study was to estimate adherence as well as the proper effect size required to calculate the

sample size for a full-scale study to assess the adherence as well as to compare two protocols of extracorporeal shockwave therapy in improving function and reducing pain in patients with planter fasciitis through a randomized controlled trial.

## Methods

### Participants

This pre-test post-test pilot randomized controlled pilot trial was performed between the seventh of November to the 20[th] of December 2023. Participants were included if they were between 20 and 50 years of age with normal Body Mass Index (BMI), had unilateral pain, persisting pain for at least 6 weeks, moderate disability as evaluated by the Foot Function Index (FFI), and pronated feet (6–9 on the Foot Posture Index). Participants were excluded from the study if they had a high BMI (above 35 kg/m$^2$), a history of surgery or fracture in the ankle or entire lower limb, a history of corticosteroid injection within the past 6 months, inability to follow or comprehend instructions, and severe foot pronation (+10 on the Foot Posture Index). This research was approved by the UoS Research Ethical Committee No: REC-22-09-27-02-S.

The main study protocol is registered on ClinicalTrials.gov Identifier: NCT06174142.

The enrollment and group adherence percentages were documented as components of this study.

Prior to the study, sample sizes were determined for the primary outcome measure, which focused on assessing pain through the Visual Analog Scale (VAS), and the secondary outcome measure, which assessed functional limitations using the Foot Function Index (FFI), based on calculations designed for pilot studies [9].

## Recruitment, randomization and allocation

### Recruitment

Individuals with plantar fasciitis were enlisted from the Alain Hospital's Physical Therapy and Rehabilitation Centre, which caters to outpatient services in the UAE. A clear explanation of the study's objectives was provided to all participants prior to commencing the assessment. Subsequently, participants provided written consent after receiving this explanation.

### Randomization

**Restricted random sampling.** This study utilized a restricted random sampling to assign participants randomly into three distinct groups. These groups were created using permuted block randomization of different sizes (3,6), ensuring an equal distribution with a ratio of 1:1:1 among the three groups (where A represents intervention 1, B represents intervention 2, and C represents the control group). Within each block, which contained a balanced number of participants from each category (A, B, and C), the order of treatments was randomly rearranged.

### Allocation and intervention

The three groups were allocated as follows (Fig 1).

### Blinding

During this pilot study, participants were blinded about the specific parameters of the used intervention.

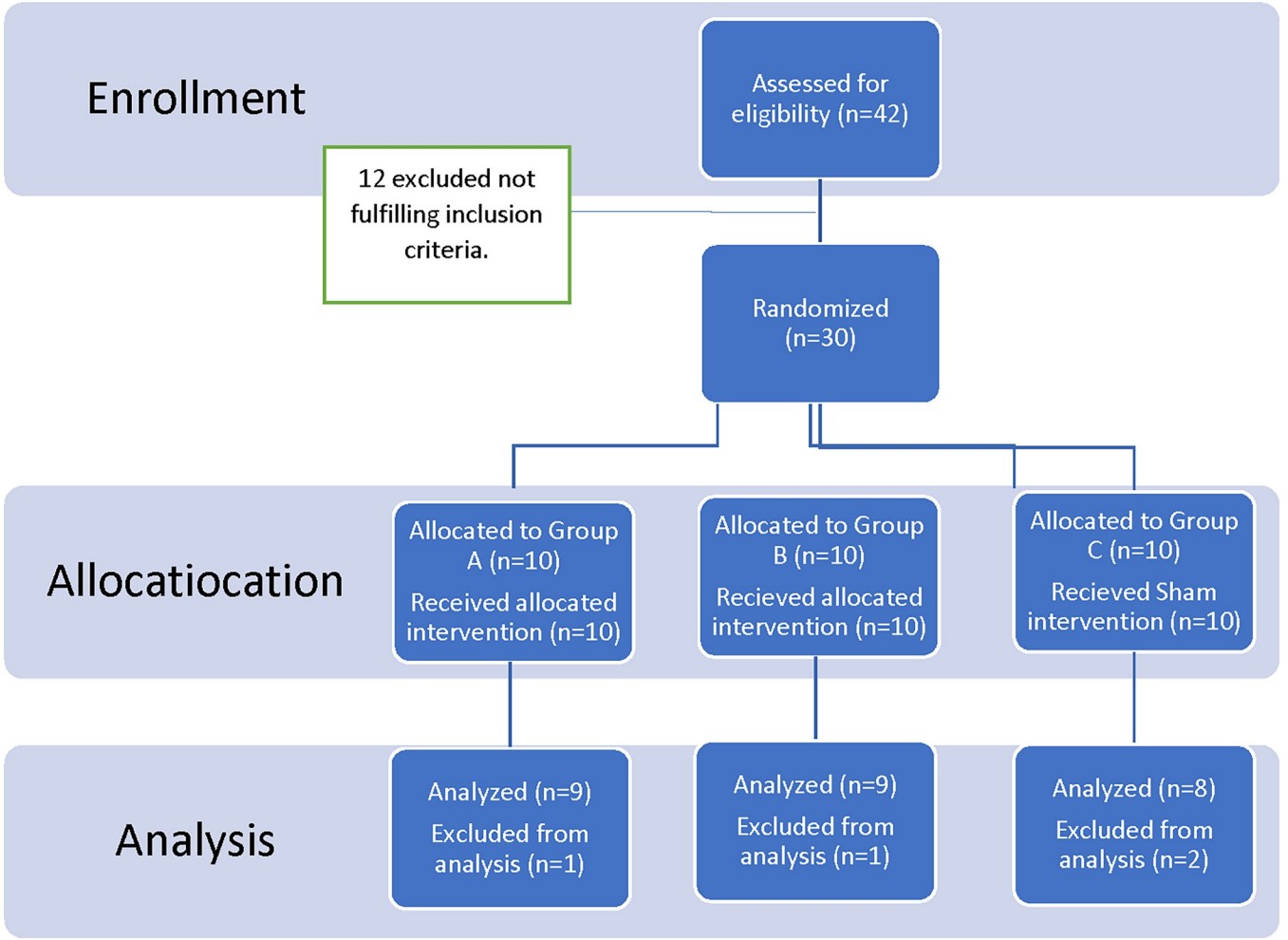

**Fig 1. Flow chart of the pilot study.** <u>Intervention Group A</u>: Patients in this category underwent a specific physical therapy regimen in conjunction with shockwave therapy (ESWT), with the following settings: Frequency: 15 Hz applied to the heel, Intensity/pressure: 3, Impulses: 1800, and a total of 6 sessions (1 session per week). <u>Intervention Group B</u>: Patients in this group also received the same designated physical therapy program along with shockwave therapy (ESWT), but with slightly different parameters: Frequency: 10 Hz on the heel, Intensity/pressure: 4, Impulses: 1800, and a total of 6 sessions (1 session per week). <u>Control Group C</u>: This group underwent the identical physiotherapy program as the intervention groups. Additionally, they received sham shockwave therapy. Pain and improvements in functionality were evaluated both before and after the 6 sessions.

### Data collection

Data collection commenced on the 7th of November 2023 and ended on the 20th of December after the participants completed the sixth session. The mean and standard deviation of pain and the FFI scores were computed at three different points in time for all participants in both groups. The initial evaluation took place prior to the intervention, the second evaluation occurred after 3 sessions, and the follow-up assessment was conducted upon completion of 6 sessions.

### Intervention

**Stretching exercise [10–12].** The stretching exercises were a core component of the intervention for all three groups—the two intervention groups and the control group. This regimen encompassed exercises targeting the plantar fascia, gastrocnemius, and soleus muscles. The

exercises were executed with the participant in a standing position, facing a wall. The participant positioned their affected foot behind the unaffected one, slightly bending the front knee. With their trunk and knee straight and the heel anchored on the floor, the participant gently leaned towards the wall until a noticeable stretch along the calf of the affected foot was felt. Each stretch was sustained for a duration of 20 to 30 seconds, and this exercise was repeated five times, for a total of five sets per day, spanning a period of six weeks.

**Extracorporeal Shockwave Therapy (ESWT) [13,14].** Extracorporeal Shockwave Therapy (Physiotur) using a 12 mm handpiece was administered to intervention group (A) with the following settings: continuous mode, a frequency of 15 Hz, pressure set at 3, and 1800 impulses. On the other hand, intervention group (B) received the therapy with different parameters: a frequency of 10 Hz, pressure set at 4, and 1800 impulses. The shockwave therapy was precisely directed to the painful area of the heel, targeting different points, including the centre and medial aspects of the heel. At each of these painful points, 200 impulses were applied until a total of 1800 impulses were achieved. In contrast, the patients in the control group (C) underwent the same physiotherapy program, supplemented with sham shockwave therapy. During the procedure, the patients were positioned in a prone posture, with both legs resting on a sigmoid wedge.

**Home exercise program.** After completing the six-week intervention, all participants were given a home exercise program designed to uphold and further improve the progress achieved during the treatment sessions. As suggested, home-based exercises tailored to address plantar fasciitis symptoms and functional enhancement have proven effective [15]. This program included the same stretching and strengthening exercises that were part of the intervention, with adjustments made to suit each individual's specific needs and abilities [11]. Participants were encouraged to perform these exercises daily for an additional six weeks, gradually increasing intensity and repetitions as their comfort allowed, following the approach [16] to assist participants in these exercises, a comprehensive exercise handout was provided, complete with step-by-step instructions and illustrations to ensure proper form and technique. This method has been demonstrated to improve adherence and comprehension [17]. Participants were also welcome to reach out to the research team if they encountered any challenges or had inquiries about their home exercise program, fostering open communication [18].

## Outcomes measures

In this study, two primary metrics were employed to evaluate the effectiveness of the interventions: the Foot Function Index (FFI) and the Visual Analog Scale (VAS).

**Foot Function Index (FFI).** The Foot Function Index (FFI) is a self-reported questionnaire comprising 23 items categorized into three domains: pain, disability, and activity limitation. Its purpose was to gauge the impact of foot pathology. Participants were instructed to rate themselves on each query, using a scale from 0 (indicating no pain or disability) to 10 (indicating the worst pain or requiring assistance). The FFI is a validated and dependable measure used in trials involving foot pathology interventions [11].

**Visual Analog Scale (VAS).** The Visual Analog Scale (VAS) is a validated assessment tool utilized to measure pain levels in both acute and chronic pain conditions. It consists of a 10 cm line that spans between two endpoints, with 0 cm representing the absence of pain and 10 cm signifying the worst pain possible. Participants were requested to mark their pain level on this line, and the recorded score reflected their pain intensity. The VAS has demonstrated satisfactory reliability for assessing acute pain [19].

## Statistical analysis

To analyse the collected data, SPSS Statistics (IBM Corp. Released 2020. IBM SPSS Statistics for Windows, Version 27.0. Armonk, NY: IBM Corp). We computed the frequency, average, standard deviation, as well as the median and interquartile ranges for the variables under examination. Data was acquired from the participants both before and after the intervention.

As data was not normally distributed, we employed the Shapiro-wilk test to check the normality of the data distribution. The Kruskal Wallis test was used to check the difference between the 3 groups. Post-hoc test was used to identify exactly which groups differ from each other, and Mann Whitney U test to check the extent of difference between the two groups that received shock wave.

## Results

The analysis involved 26 participants categorized into three groups: Group A, Group B, and Group C, each comprising 9,9 and 8 individuals respectively. In Group A, participants received ESWT with a frequency of 15 Hz and intensity of 3, while Group B received ESWT with a frequency of 10 Hz and intensity of 4. Group C, established as the control group, underwent the designated physiotherapy program alongside sham shock wave therapy.

Adherence rates were reported high (90%, 90% and 80% respectively) in all three groups.

Adherence was calculated by dividing the number of participants completing the study by the total number of each group.

Evaluations of pain and functional improvement were conducted before and after six sessions.

Baseline variables (Table 1), including age, gender, and BMI, were taken into consideration. Based on the small sample size, the median and inter quartile ranges (IQR) were considered. The median ages and IQRs of participants in Group A, Group B, and Group C were 36.5 (15), 43.0 (9.8), and 31.5 (12.3) years. The gender distribution in each group was comparable, with Group A consisting of 66.7% females and 33.3% males, Group B comprising 66.7% females and 33.3% males, and Group C containing 62.5% females and 37.5% males.

The median BMI of participants in the three groups exhibited similarity, with values of 24.3, 24.6, and 23.5, respectively. Pain and functional improvement were assessed using the Visual Analog Scale (VAS) and Foot Function Index (FFI), respectively. Initial VAS scores were the same across all groups, with a median score of 6.

**Table 1. Variable characteristics.**

| Variables | Group A | Group B | Group C |
|---|---|---|---|
| **Age,** Years, Median (IQR) | 36.5 (15) | 43.0 (9.8) | 31.5 (12.3) |
| **Gender,** Count (%) | | | |
| Female | 6 (66.7%) | 6 (66.7%) | 5 (62.5%) |
| Male | 3 (33.3%) | 3 (33.3%) | 3 (37.5%) |
| **BMI kg/m$^2$** | 24.3 | 24.6 | 23.5 |
| **VAS initial,** Median (IQR) | 6 | 6 | 6 |
| **VAS 3rd session,** Median (IQR) | 4 (1) | 4 (2) | 5.5 (1) |
| **VAS 6th session,** Median (IQR) | 0.5 (1) | 1.5 (1.3) | 4 (1) |
| **FFI initial,** Median (IQR) | 0.48 (0) | 0.47 (9.25) | 0.50 (10) |
| **FFI 3rd session,** Median (IQR) | 0.30 (9) | 0.31 (17.5) | 0.45 (10.25) |
| **FFI 6th session,** Median (IQR) | 0.04 (6.4) | 0.11 (8.75) | 0.32 (9) |

**Table 2. Kruskal-Wallis test; comparison between groups.**

| | VAS initial | VAS 4th session | VAS 7 session | FFI initial | FFI 4th session | FFI 7 session |
|---|---|---|---|---|---|---|
| | | | Test Statistics[a] | | | |
| Chi-Square | 1.736 | 17.077 | 21.677 | 1.876 | 17.928 | 22.071 |
| df | 2 | 2 | 2 | 2 | 2 | 2 |
| Asymp. Sig. | .420 | .000 | .000 | .391 | .000 | .000 |

a. Kruskal Wallis Test.

*Significant (p<0.05).

Following the third session, VAS scores in Group A and Group B decreased to 4, indicating an improvement in pain, while Group C's VAS score remained relatively high at 5.5. By the end of the sixth session, VAS scores in Group A and Group B further reduced to 0.5 and 1.5, signifying significant pain reduction, whereas Group C's score only decreased to 4.

Initial FFI scores were also similar among the three groups, with values of 0.48for Group A and 0.47 for Group B, and 0.50 for Group C. After the third session, FFI scores in Group A and Group B decreased to 0.30 and 0.31 respectively, while Group C's score was only reduced to 0.45. By the sixth session, FFI scores in Group A and Group B further decreased to 0.04 and 0.11, respectively, indicating substantial functional improvement. In contrast, although decreased, Group C's score remained higher at 0.32.

Comparing groups, there were no significant differences between the groups in the **initial** session in terms of VAS and FFI (P>0.05), however, it was noted that there was a significant decrease in VAS and FFI after the 3rd and 6th sessions (P<0.001) (Table 2).

In addition, pairwise comparisons (Table 3) revealed no differences between groups A and B in terms of VAS and FFI scores.

With regards to the Post hoc analysis, results of the 3rd session showed similar scores in group A and B, compared to group C.

**Table 3. Pairwise comparisons between groups A, B and C.**

| Sample1-Sample2 | Test Statistic | Std. Error | Std. Test Statistic | Sig. | Adj. Sig. |
|---|---|---|---|---|---|
| | | | VAS at 3rd session | | |
| A-B | -1.600 | 3.792 | -.422 | .673 | 1.000 |
| A-C | -14.300 | 3.792 | -3.771 | .000 | .000* |
| B-C | -12.700 | 3.792 | -3.349 | .001 | .002* |
| | | | VAS at 6th session | | |
| A-B | -4.700 | 3.855 | -1.219 | .223 | .668 |
| A-C | -17.350 | 3.855 | -4.501 | .000 | .000* |
| B-C | -12.650 | 3.855 | -3.282 | .001 | .003* |
| | | | FFI at 3rd session | | |
| A-B | -4.000 | 3.920 | -1.020 | .308 | .923 |
| A-C | -15.950 | 3.920 | -4.069 | .000 | .000* |
| B-C | -11.950 | 3.920 | -3.049 | .002 | .007* |
| | | | FFI at 6th session | | |
| A-B | -6.200 | 3.916 | -1.583 | .113 | .340 |
| A-C | -18.100 | 3.916 | -4.622 | .000 | .000* |
| B-C | -11.900 | 3.916 | -3.039 | .002 | .007* |

* Significant (p<0.05).

Meanwhile, results of the 6th session showed that group A was superior to groups B and C, with group B displaying lower scores in terms of VAS and FFI when compared to group C (Table 3).

For future sample size calculations, considering the VAS as the primary outcome, the effect size for post-hoc Kruskal-Wallis test was calculated using the formula $r = z/\sqrt{N}$ (r: effect size; z: z value; N: Observation number).

The effect sizes between the groups were reported as 0.282, 1.09 and 0.796 respectively.

A total sample size of 123 participants was calculated using the G-power software for the ANOVA repeated measures within-between interaction test, assuming an effect size of 0.282, α level of 0.05, and power of 0.95.

## Discussion

The aim of the study was to detect adherence rates as well as the effect size required to estimate the actual sample size needed for a larger scale study to compare the effectiveness of two different extracorporeal shock wave therapy (ESWT) protocols combined with a specified physical therapy program, in relieving pain and improving foot function in individuals with plantar fasciitis. In addition, the study aimed to report the effects of the ESWT protocols used on pain and function.

The study involved evaluating outcomes using the Visual Analog Scale (VAS) and Foot Function Index (FFI) and included a comparative analysis between ESWT administered at two distinct frequency and intensity levels and a sham therapy group. The results of our study indicate that the utilization of shockwave therapy (ESWT) in combination with a specific physical therapy program has notable impacts on decreasing pain and enhancing functionality in individuals diagnosed with plantar fasciitis.

The results indicated a high adherence rate for all groups. One reason to investigate adherence or compliance is the absence of definitions and standards for satisfactory compliance within research which are considered methodological shortcomings [20]. Subjects' willingness to volunteer for studies of the effects of investigational agents [21] and commitment in following treatment protocols are fundamental to clinical research. However, once enrolled in a clinical study, some participants fail to comply [22].

Our results came within the accepted limits of compliance as a convention in biomedical research known as the "80% rule" has been used as an operational criterion for regimen adherence [23]. Another important reason that we needed to avoid in later is the fact that scientific journals are reluctant to publish articles with lower adherence rates [24].

The study also reported a small effect size detected of 0.25 that would suggest a total of 120 participants needed to replicate the study on a larger scale.

Although this was a pilot study, still findings were consistent with previous literature, where significant improvement was observed in both intervention groups (A and B) compared to the control group, suggesting that ESWT is an effective treatment modality for plantar fasciitis [25]. Although both intervention groups exhibited notable enhancements in pain (VAS) and function (FFI) scores after the third and sixth sessions, Group A displayed slightly superior outcomes compared to Group B during the sixth session. This discrepancy may be attributed to the differences in frequency and intensity employed in the two ESWT protocols. Group A utilized a higher frequency (15 Hz) and lower intensity (pressure 3), whereas Group B employed a lower frequency (10 Hz) and higher intensity (pressure 4). It was previously noted that higher frequencies are more effective in treating plantar fasciitis, this supports our results

[26] when considering the physiological implications of high frequencies, it is essential to comprehend how ESWT influences the body on a cellular level. This is due to the fact that plantar fasciitis is a biomechanical overuse syndrome resulting in degenerative changes at its attachment to the calcaneus. Histologic examination of samples taken from plantar fascia release surgeries revealed myxoid degeneration with fragmentation and degeneration of the plantar fascia and bone marrow vascular ectasia [2].

Specifically, at higher frequencies, extracorporeal shock wave therapy operates by triggering a biological response that supports tissue repair. Mechano-transduction, the conversion of mechanical stimuli into biochemical signals, is a pivotal factor in this mechanism. Higher frequency shock waves can heighten cellular activity by effectively stimulating more mechanoreceptors in the targeted tissue, thus amplifying the healing response. The heightened frequency of mechanical stimuli facilitates the release of growth factors, promotes the formation of blood vessels (angiogenesis), and encourages cell proliferation and differentiation, collectively contributing to enhanced healing and regeneration of the plantar fascia [27–30]. These biological responses caused by higher frequency shock waves could provide an explanation for the superior results observed in Group A, suggesting that higher frequency ESWT could be more advantageous for individuals with plantar fasciitis. Future studies are required to investigate the intricate cellular and molecular mechanisms through which variations in frequency impact the efficacy of ESWT.

Moreover, the notable variations in VAS and FFI scores between the intervention groups and the control group emphasize the significance of incorporating ESWT into the treatment regimen. These findings are consistent with prior studies that have demonstrated the efficacy of ESWT in addressing plantar fasciitis [14]. The chosen physical therapy program also contributed to enhancing patients' functional outcomes, given the well-established benefits of physical therapy in addressing musculoskeletal pain and dysfunction [31].

However, to enhance generalizability and validate these results, it is imperative to emphasize the necessity of conducting a comprehensive full-scale study subsequent to this pilot investigation. A larger study would allow for broader participant inclusion and further inspection of the observed outcomes.

## Conclusion

This pilot study provided a high adherence rate for the three groups as well as a small effect size detected of 0.282 that would suggest a total of 123 (at alpha level of 0.05 and power of 0.95) participants required to replicate the study on a larger scale.

In addition, it provided preliminary evidence that combining ESWT with a specific physical therapy regimen for treating plantar fasciitis is effective. The findings suggest that the ESWT protocol featuring a higher frequency and lower intensity leads to slightly superior outcomes in both pain reduction and functional improvement compared to the protocol with a lower frequency and higher intensity.

Clinicians could be advised to consider integrating ESWT into their treatment strategies for patients with plantar fasciitis, adjusting the frequency and intensity to achieve optimal results. However, this must be confirmed with larger sample sizes and extended follow-up periods is necessary to validate these results and investigate the optimal ESWT parameters for effectively managing plantar fasciitis.

## Supporting information

**S1 Checklist. CONSORT 2010 checklist of information to include when reporting a pilot or feasibility trial\*.**
(DOC)

**S1 File.**
(PDF)

## Author Contributions

**Conceptualization:** Fatima A. L. Kalbani, Tamer Shousha.

**Data curation:** Tamer Shousha.

**Formal analysis:** Tamer Shousha.

**Investigation:** Fatima A. L. Kalbani, Reime Shalash, Raneen Qadah, Tamer Shousha.

**Methodology:** Fatima A. L. Kalbani, Tamer Shousha.

**Project administration:** Tamer Shousha.

**Software:** Reime Shalash, Raneen Qadah.

**Supervision:** Tamer Shousha.

**Validation:** Tamer Shousha.

**Visualization:** Tamer Shousha.

**Writing – original draft:** Fatima A. L. Kalbani, Reime Shalash, Raneen Qadah, Tamer Shousha.

**Writing – review & editing:** Tamer Shousha.

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
