## [Decision Letter · Decision Letter 0]

4 Feb 2024

PONE-D-23-43583Comparing Two Protocols of Shock Wave Therapy for Patients with Plantar Fasciitis: A Pilot StudyPLOS ONE

Dear Dr. Shousha,

Thank you for submitting your manuscript to PLOS ONE. After careful consideration, we feel that it has merit but does not fully meet PLOS ONE’s publication criteria as it currently stands. Therefore, we invite you to submit a revised version of the manuscript that addresses the points raised during the review process.

We look forward to receiving your revised manuscript.

Kind regards,

Filippo Migliorini

Academic Editor

PLOS ONE

Additional Editor Comments:

Reviewer 1:

The authors present a meta-analysis comparing and comparing the effectiveness of two extracorporeal shock wave therapy (ESWT) protocols combined with a specified physical therapy program in relieving pain and improving foot function in individuals with plantar fasciitis. This topic is of great clinical

interest and has been extensively discussed in the literature. The analysis itself is very thoroughly performed and clearly explained. The discussion is precious, and I also appreciate the explanation of your way of understanding and performing the study. However, some minor and major corrections should be addressed.

L61-62: Does it cause only heel pain?

Materials and methods:

It has not been specified if you followed a guideline and, in this case, which one.

Discussion:

L306: A short description of the anatomical structures and the pathological alterations are needed before talking about the impact of the shock waves on cells.

Conclusion:

no comments

Reviewers' comments:

Reviewer's Responses to Questions

**Comments to the Author**

1. Is the manuscript technically sound, and do the data support the conclusions?

Reviewer #1: Yes

2. Has the statistical analysis been performed appropriately and rigorously? 

Reviewer #1: I Don't Know

3. Have the authors made all data underlying the findings in their manuscript fully available?

Reviewer #1: Yes

4. Is the manuscript presented in an intelligible fashion and written in standard English?

Reviewer #1: Yes

5. Review Comments to the Author

Reviewer #1: The authors present a meta-analysis comparing and comparing the effectiveness of two extracorporeal shock wave therapy (ESWT) protocols combined with a specified physical therapy program in relieving pain and improving foot function in individuals with plantar fasciitis. This topic is of great clinical

interest and has been extensively discussed in the literature. The analysis itself is very thoroughly performed and clearly explained. The discussion is precious, and I also appreciate the explanation of your way of understanding and performing the study. However, some minor and major corrections should be addressed.

L61-62: Does it cause only heel pain?

Materials and methods:

It has not been specified if you followed a guideline and, in this case, which one.

Discussion:

L306: A short description of the anatomical structures and the pathological alterations are needed before talking about the impact of the shock waves on cells.

Conclusion:

no comments

6. PLOS authors have the option to publish the peer review history of their article (what does this mean?). If published, this will include your full peer review and any attached files.

Reviewer #1: No

---

## [Author Response · Author response to Decision Letter 0]

7 Feb 2024

Dear Reviewer, 

Thank you for your thorough revision and helpful comments…

Kindly find below, the authors’ reply on a point-to-point basis.

Kindly 

Comment Authors’ reply

L61-62: Does it cause only heel pain? Added lines 62-64: ‘, with symptoms of stabbing, non-radiating pain in the early morning of the proximal medio-plantar surface of the foot; the pain becomes worse at the end of the day [2].”

Materials and methods:

It has not been specified if you followed a guideline and, in this case, which one As this is a pilot study, we followed the “CONSORT 2010 checklist of information to include when reporting a pilot or feasibility trial” which is already one of the journal’s requirements for submission and was uploaded with the supporting files.

In addition, the sample calculation was done in accordance to the study “Hertzog MA. Considerations in determining sample size for pilot studies. Res Nurs Health. 2008;31: 180–191” reference 9 stated in line 115.

All assessment and treatment parameters have references included in text

L306: A short description of the anatomical structures and the pathological alterations are needed before talking about the impact of the shock waves on cells. Added lines 308-312(version without track changes) / 313-317( version with track changes): “This is due to the fact that plantar fasciitis is a biomechanical overuse syndrome resulting in degenerative changes at its attachment to the calcaneus. Histologic examination of samples taken from plantar fascia release surgeries revealed myxoid degeneration with fragmentation and degeneration of the plantar fascia and bone marrow vascular ectasia [2].”

---

## [Decision Letter · Decision Letter 1]

17 Mar 2024

PONE-D-23-43583R1Comparing Two Protocols of Shock Wave Therapy for Patients with Plantar Fasciitis: A Pilot StudyPLOS ONE

Dear Dr. Shousha,

Thank you for submitting your manuscript to PLOS ONE. After careful consideration, we feel that it has merit but does not fully meet PLOS ONE’s publication criteria as it currently stands. Therefore, we invite you to submit a revised version of the manuscript that addresses the points raised during the review process.

We look forward to receiving your revised manuscript.

Kind regards,

Filippo Migliorini

Academic Editor

PLOS ONE

Journal Requirements:

Additional Editor Comments:

The article has been evaluated by a statistician. Please revise the manuscript accordingly, thank you

Comments:

This manuscript reported results from a pilot randomized control trial with three arms: two different shock wave therapy and one control. There are 30 patients randomized in this trial. In order to meet the standard for publication, the statistical analysis part needs revisions and clarification as shown below:

1. What is the block size used in the permuted block randomization? This information should be added.

2. The follow chart shows the final number of patients analyzed was 9, 9 and 8 for three groups. But were all 10 patients' baseline measurements analyzed and shown in Table 1? It is not clear how compliance was defined.

3. Because of the small sample size in each group, median+/-IQR should also be reported. Because of the same reason, Shapiro-Wilks test is not appropriate for testing the normality assumption.

4. Table 3 reports the comparison between group A and group B as well as the first row in Table 4. First, such information is duplicated so Table 3 could be removed. Second, It is confusing that p-values from the same comparisons in these two tables were not the same.

5. Line 251 and Line 269 have duplicated information presented.

6. There are longitudinal measurements observed from each study participants. It is better to draw individual profile of each outcome over time to show the difference visually. More importantly, the statistical analysis such as linear mixed effect model is better to take advantage of such longitudinal design for estimating the difference between groups, which would be more powerful. Such model can also use all available observed data even if some study participants only had baseline measurements recorded.

7. There should be more details about future sample size calculation. For example, which group difference was the effect size of 0.25 from, group A vs group C, group B vs group C, or from all three groups? How was the effect size calculated? Which outcome was used in that effect size calculation, FFI or VAS? Which statistical test was used in this sample size calculation? What is the alpha and beta level used in such calculation?

Reviewers' comments:

Reviewer's Responses to Questions

**Comments to the Author**

1. If the authors have adequately addressed your comments raised in a previous round of review and you feel that this manuscript is now acceptable for publication, you may indicate that here to bypass the “Comments to the Author” section, enter your conflict of interest statement in the “Confidential to Editor” section, and submit your "Accept" recommendation.

Reviewer #2: All comments have been addressed

2. Is the manuscript technically sound, and do the data support the conclusions?

Reviewer #2: Partly

3. Has the statistical analysis been performed appropriately and rigorously? 

Reviewer #2: No

4. Have the authors made all data underlying the findings in their manuscript fully available?

Reviewer #2: Yes

5. Is the manuscript presented in an intelligible fashion and written in standard English?

Reviewer #2: Yes

6. Review Comments to the Author

Reviewer #2: This manuscript reported results from a pilot randomized control trial with three arms: two different shock wave therapy and one control. There are 30 patients randomized in this trial. In order to meet the standard for publication, the statistical analysis part needs revisions and clarification as shown below:

1. What is the block size used in the permuted block randomization? This information should be added.

2. The follow chart shows the final number of patients analyzed was 9, 9 and 8 for three groups. But were all 10 patients' baseline measurements analyzed and shown in Table 1? It is not clear how compliance was defined.

3. Because of the small sample size in each group, median+/-IQR should also be reported. Because of the same reason, Shapiro-Wilks test is not appropriate for testing the normality assumption.

4. Table 3 reports the comparison between group A and group B as well as the first row in Table 4. First, such information is duplicated so Table 3 could be removed. Second, It is confusing that p-values from the same comparisons in these two tables were not the same.

5. Line 251 and Line 269 have duplicated information presented.

6. There are longitudinal measurements observed from each study participants. It is better to draw individual profile of each outcome over time to show the difference visually. More importantly, the statistical analysis such as linear mixed effect model is better to take advantage of such longitudinal design for estimating the difference between groups, which would be more powerful. Such model can also use all available observed data even if some study participants only had baseline measurements recorded.

7. There should be more details about future sample size calculation. For example, which group difference was the effect size of 0.25 from, group A vs group C, group B vs group C, or from all three groups? How was the effect size calculated? Which outcome was used in that effect size calculation, FFI or VAS? Which statistical test was used in this sample size calculation? What is the alpha and beta level used in such calculation?

7. PLOS authors have the option to publish the peer review history of their article (what does this mean?). If published, this will include your full peer review and any attached files.

Reviewer #2: No

---

## [Author Response · Author response to Decision Letter 1]

27 Mar 2024

Dear Reviewer, 

Thank you for your thorough revision and helpful comments…

Kindly find below, the authors’ reply on a point-to-point basis.

Kindly note that the line numbers mentioned below are in the copy with track changes

Warm regards,

Comment Authors’ reply

What is the block size used in the permuted block randomization? Thank you for your comment. 

Lines 125 -126: added “permuted block randomization of different sizes (3,6)”

The flow chart shows the final number of patients analyzed was 9, 9 and 8 for three groups. But were all 10 patients' baseline measurements analyzed and shown in Table 1? It is not clear how compliance was defined. Thank you for your comment. The percentages were checked, and correct percentages were edited in text lines: 230-232 as well as in Table 1

Compliance/ adherence was calculated by dividing the number of participants completing the study by the total number of each group . lines 226-227

Because of the small sample size in each group, median+/-IQR should also be reported. Because of the same reason, Shapiro-Wilks test is not appropriate for testing the normality assumption. 

Thank you for your comment. 

Median and IQR values added to Table 1

With regards to the Shapiro-wilk test, the choice was based on previous literature revealing the Shapiro-wilk appropriate for sample sizes less than 50.

Elliott AC, Woodward WA. Statistical analysis quick reference guidebook with SPSS examples. 1st ed. London: Sage Publications; 2007

Mishra P, Pandey CM, Singh U, Gupta A, Sahu C, Keshri A. Descriptive statistics and normality tests for statistical data. Ann Card Anaesth. 2019 Jan-Mar;22(1):67-72. doi: 10.4103/aca.ACA_157_18. PMID: 30648682; PMCID: PMC6350423.

Ghasemi A, Zahediasl S. Normality tests for statistical analysis: a guide for non-statisticians. Int J Endocrinol Metab. 2012 Spring;10(2):486-9. doi: 10.5812/ijem.3505. Epub 2012 Apr 20. PMID: 23843808; PMCID: PMC3693611.

Table 3 reports the comparison between group A and group B as well as the first row in Table 4. First, such information is duplicated so Table 3 could be removed. Second, It is confusing that p-values from the same comparisons in these two tables were not the same. The table has been removed and replaced with the pairwise comparisons of the KW test

Line 251 and Line 269 have duplicated information presented Duplication removed

There are longitudinal measurements observed from each study participants. It is better to draw individual profile of each outcome over time to show the difference visually. More importantly, the statistical analysis such as linear mixed effect model is better to take advantage of such longitudinal design for estimating the difference between groups, which would be more powerful. Such model can also use all available observed data even if some study participants only had baseline measurements recorded Thank you for this comment…

The linear mixed effect model depends on the assumption for normality which is not available in our case since the data was not normally distributed.

 There should be more details about future sample size calculation. For example, which group difference was the effect size of 0.25 from, group A vs group C, group B vs group C, or from all three groups? How was the effect size calculated? Which outcome was used in that effect size calculation, FFI or VAS? Which statistical test was used in this sample size calculation? What is the alpha and beta level used in such calculation? Thank you for your valuable comment…

Added lines 288-292: For future sample size calculations, considering the VAS as the primary outcome, the effect size for post-hoc Kruskal-Wallis test was calculated using the formula r = z/√N (r: effect size; z: z value; N: Observation number).

The effect sizes between the groups were reported as 0.282, 1.09 and 0.796 respectively.

A total sample size of 123 participants was calculated using the G-power software for the ANOVA repeated measures within-between interaction test, assuming an effect size of 0.282, α level of 0.05, and power of 0.95.

---

## [Editor Report · Decision Letter 2]

8 Apr 2024

Comparing Two Protocols of Shock Wave Therapy for Patients with Plantar Fasciitis: A Pilot Study

PONE-D-23-43583R2

Dear Dr. Shousha,

We’re pleased to inform you that your manuscript has been judged scientifically suitable for publication and will be formally accepted for publication once it meets all outstanding technical requirements.

Kind regards,

Filippo Migliorini

Academic Editor

PLOS ONE
---

## [Editor Report · Acceptance letter]

26 Apr 2024

PONE-D-23-43583R2 

PLOS ONE

Dear Dr. Shousha, 

I'm pleased to inform you that your manuscript has been deemed suitable for publication in PLOS ONE. Congratulations! Your manuscript is now being handed over to our production team.

Kind regards, 

on behalf of

Dr Filippo Migliorini 

Academic Editor

PLOS ONE